# Effects of Sprinkler Flow Rate on Physiological, Behavioral and Production Responses of Nili Ravi Buffaloes during Subtropical Summer

**DOI:** 10.3390/ani11020339

**Published:** 2021-01-29

**Authors:** Musa Bah, Muhammad Afzal Rashid, Khalid Javed, Talat Naseer Pasha, Muhammad Qamer Shahid

**Affiliations:** 1Department of Livestock Management, University of Veterinary and Animal Sciences, Lahore 54000, Pakistan; mmbah@utg.edu.gm (M.B.); khalidjaved@uvas.edu.pk (K.J.); 2School of Agriculture and Environmental Sciences, University of The Gambia, Serekunda 3530, Republic of Gambia; 3Department of Animal Nutrition, University of Veterinary and Animal Sciences, Lahore 54000, Pakistan; drafzal@uvas.edu.pk (M.A.R.); tnpasha@uvas.edu.pk (T.N.P.)

**Keywords:** behavior, dairy buffaloes, heat stress, performance, sprinkler cooling, welfare

## Abstract

**Simple Summary:**

Water buffaloes wallow in water to combat heat stress during summer. With the decreasing reservoirs for wallowing, the farmers use hosepipes and sprinklers to cool the buffaloes in Pakistan. The sprinklers use a large quantity of groundwater, which is becoming scarce. In the current study, different flow rates in sprinklers were evaluated as a strategy to optimally use groundwater to cool water buffaloes. The sprinkler flow rates 1.25 and 2.0 L/min both had better physiological response, more feed intake, and higher milk yield compared to the 0.8 L/min. However, the flow rate 1.25 L/min was more efficient in cooling the buffaloes because it yielded similar physiological, production, and the behavioral responses despite using 37.5% less groundwater compared to 2.0 L/min. The current findings will help farmers to reduce the amount of groundwater used in cooling dairy buffaloes.

**Abstract:**

Water buffaloes wallow in water to combat heat stress during summer. With the decreasing reservoirs for wallowing, the farmers use sprinklers to cool the buffaloes in Pakistan. These sprinklers use a large quantity of groundwater, which is becoming scarce. The objective of the current study was to determine the effect of different sprinkler flow rates on the physiological, behavioral, and production responses of Nili Ravi buffaloes during summer. Eighteen buffaloes were randomly subjected to three sprinkler flow rate treatments in a double replicated 3 × 3 Latin square design. The flow rates were 0.8, 1.25, and 2.0 L/min. During the study, the average afternoon temperature humidity index was 84.6. The 1.25 and 2.0 L/min groups had significantly lower rectal temperature and respiratory rates than the 0.8 L/min group. Water intake was significantly higher in the 0.8 L/min group. Daily milk yield was higher in the 1.25 and 2.0 L/min groups than in the 0.8 L/min group. These results suggested that the sprinkler flow rates > 0.8 L/min effectively cooled the buffaloes. The sprinkler flow rate of 1.25 L/min appeared to be more efficient, as it used 37.5% less water compared to the 2.0 L/min.

## 1. Introduction

Water buffaloes (*Bubalus bubalis*) are the major dairy animals in the Indo-Pak subcontinent [1], where heat stress is a main challenge for livestock during summer. Water buffaloes are adapted to hot and humid environments and rely on wallowing in water to abate heat load [2]. The changes in buffalo farming systems [3] and reduction in potholes and ponds in villages has led to limited or no access to water for wallowing. Alternatively, the farmers use hosepipes and sprinklers to cool the buffaloes in Pakistan. These cooling methods use a large quantity of groundwater, which is becoming scarce.

The most common method of cooling dairy animals during summer is the use of sprayed water from sprinklers/soakers or misters [4,5]. Compared with the shade alone, sprayed water reduces body temperature [6,7,8], respiration rate [7,8,9], improves lying time, feed intake [7,10], and milk yield [7] in cows. Studies on the buffaloes also indicated that the provision of sprinklers lowered respiration rate and body temperature, and increased dry matter intake and milk yield compared to non-cooled buffaloes during summer [11,12,13]. However, a large quantity of groundwater is used in spraying cattle at dairy farms [7], which raises concerns of water footprints and water-use efficiency [14]. In our preliminary survey, we found that farmers in the rural and peri-urban areas of Punjab, Pakistan, used up to 350 L of groundwater to cool a single buffalo per day during summer, which was higher than the water used for drinking and cleaning (unpublished data). Pakistan is projected to become water scarce in 2035 [15], hence the judicious use of this paramount resource is very important. Declining levels of groundwater due to climate change [16] makes the water-use efficiency an important issue for sustainable livestock production across the globe.

In recent years, few studies have evaluated the effectiveness of different spray water saving strategies in cooling dairy cows. Some evaluated the sprinkler flow rate [5,7], others the duration of the spray [8,16], and droplet size [9,17]. In a hot, dry Mediterranean climate, the cows housed in a group pen cooled with sprinklers having 1.3 L/min flow rate had better physiological responses compared to 0.4 L/min [18]. However, increasing sprinkler flow to ≥4.5 L/min did not improve the physiological responses relative to 1.3 L/min of these cows [18]. Likewise, an earlier study showed no additional benefits of cooling cows with higher flow rates (8.2 and 11.7 L/min) compared to the lowest (5.2 L/min in a freestall barn) [19].

To our knowledge, no spray water saving strategies in cooling dairy buffaloes have been investigated. The objective of our study was to determine the effect of different sprinkler flow rates on the physiological, production, and behavioral responses of Nili Ravi buffaloes during summer in Pakistan.

## 2. Materials and Methods

### 2.1. Animals, Housing, and Management

The present study was conducted at the Dairy Animals Training and Research Center, the University of Veterinary and Animal Sciences (UVAS), Lahore, Ravi Campus, Pattoki, Pakistan (31°03′43.9″ N 73°52′36.1″ E) during summer (May to June 2019).

Eighteen lactating Nili Ravi buffaloes were used, with average daily milk yield 5.7 ± 1.4 Kg, days in milk 92.4 ± 54.2, parity 3.6 ± 2.2, and body weight 552.6 ± 84.7 Kg (mean ± SD). The buffaloes were housed in a naturally ventilated shed with asbestos roofing and a concrete floor. The shed was about 30 m long (east-west) and 12 m wide with a centrally located manger. A polyvinyl water pipe was installed along the feed manger about 2.3 m above the floor. The sprinklers were fitted with a solenoid valve on the water pipe at a distance of 2.25 m apart and about 49 cm away from the proximal edge of the manger. The sprinkler nozzles had 180-degree radius and angled to spray water at the back of the buffaloes. The side walls of the shed had industrial fans (Model FS-75, Bilal Electronics, Lahore, Pakistan; blade length 36.5 cm, width 6.5 cm;) mounted at a height of 2.4 m blowing towards the buffaloes. A schematic diagram of the shed is presented in Figure 1.

The buffaloes were tied at the manger with neck chains during the daytime (8:00 to 1800 h) and released in the adjacent outdoor open area during the nighttime. The distance between the buffaloes was similar to that of sprinkler nozzles (2.25 m) to have one sprinkler for one buffalo. The outdoor open area had sand as a bedding material. The release of buffaloes to the outdoor open area was done because there was no showering during the nighttime and to provide a comfortable resting surface during the night. Individual water tubs were placed for each tied buffalo in the shed during the daytime and water trough in the open area was available for free access during the nighttime. Milking was done in a 6 × 6 herringbone milking parlor (GEA Farm Technologies, Bönen, Germany) around 5:00 and 17:00 h daily.

### 2.2. Experimental Design

Three sprinkler flow rate treatments were tested on 18 buffaloes in a double replicated 3 × 3 Latin square design. The flow rates were 0.8, 1.25, and 2.0 L/min. Nine buffaloes with nine sprinklers (1 sprinkler nozzle/buffalo) were used in square 1, and the other nine in square 2. The treatments were applied from 8:00 to 18:00 h/d. Each square consisted of 3 periods with 7 d per period. Each treatment was applied on 3 animals in a period and repeated on three other animals in each subsequent period, making it 9 experimental units per treatment in a single square as shown in Table 1. The treatments were applied in a similar way in square 2 on the other 9 buffaloes. The double square was used to increase the number of experimental units for better reliability of the study outcome. The first square took place from 16 May to 5 June, and the 2nd square from 10 June to 30 June. The sprinklers and fans were turned on manually from 8:00 until 18:00 h in a 12-min cycle with 3 min water on, followed by 9 min off [7]. The 2.0 L/min flow rate was chosen as it was used by the traditional dairy industry in the study area. The other two treatments were chosen as a strategy to reduce water use for cooling dairy animals. At the start, the lowest flow rate of 0.5 L/min was tested, but it did not create sprinkling, rather it was just dripping. The minimum water flow rate of 0.8 L/min produced sprinkling with the droplet size almost similar to that of raindrops and was selected as the lowest flow rate treatment. The solenoid valves were used to adjust the water flow rates. The different sprinkler flow rate characteristics are shown in Table 2. The first three nozzles were grouped as 1, the next three as 2, and the last three as 3. The application of sprinkler flow rates was randomly applied on these groups. The distance between the buffaloes was enough not to have showering effect from the adjacent nozzles. The first 4 days of each period were taken as an adaptation and the data were collected during the remaining three days.

### 2.3. Meteorological Measures

A digital thermo-humidity meter (HTC1, China) was used to record the temperature (T, °C) and relative humidity (RH, %) of the shed and the outdoor open area. The readings were taken at four different time points (6:00, 13:00, 15:00, and 18:00 h) at a height of 2.3 m in the middle of the shed. The averages of 13:00 and 15:00 h were presented as afternoon category, 6:00 h as morning, and 18:00 h as evening. The same meteorological measures were recorded at the same height and time in the outdoor open area. The following equation was used to calculate the temperature humidity index (THI; [20]):THI = (1.8 × T °C + 32) – [(0.55 – 0.0055 × RH) × (1.8 × T °C − 26)]

### 2.4. Production Measures

Feeding was done according to the farm practices. A measured quantity of total mixed ration (TMR) was offered once daily between 8:00 to 9:00 h. The TMR consisted of oat silage (91%), corn grains (4%), canola meal (4%), and molasses (1%). Due to limited resources of the farm, the inclusion level of concentrates was kept low. The orts were collected on the following morning. Feed intake was obtained by subtracting orts from feed offered. Feed samples for each period were collected, and dry matter (DM) was determined by drying samples in a hot air oven (UN260-Memmert, Schwabach, Germany) for 3 h at 105 °C.

The water intake (L) was measured only for the daytime hours (8:00 to 18:00 h) per animal. The water tubs were filled thrice during the day at 9:00, 12:00, and 15:00 h with a known quantity, and the intake was determined by subtracting the remaining water at 18:00 h. The level of water in the tubs was maintained about 3 cm below the upper edge during each filling to avoid water wastage due to splashing and feet dipping. During the night, the buffaloes were in the outdoor open area as a group hence individual water intake was not recorded. The milk yield was recorded in kilograms from the milk meters installed in the milking parlor during each milking. The daily milk yield was the sum of the morning and the evening milking.

### 2.5. Physiological Measures

Rectal temperature (RT) and respiration rate (RR) were taken between 13:00 and 14:00 h. The RT was taken using a digital thermometer (Certeza, FT-707, Hamburg, Germany) that was inserted in the rectum of a humanely restrained buffalo (using a halter and grooming the neck) and maintained until the beep sound was heard and the digits had stopped blinking. The readings (degrees Fahrenheit) were converted to degrees Celsius and rounded up to one decimal point. The RR was taken by counting the flank movements of the abdomen for 30 s timing using a stop watch on a mobile phone. The results were converted to breaths per minute by multiplying the total by 2. These conversions were made to facilitate comparisons with other studies. The parameters were recorded by the same person throughout the experimental period to avoid bias.

### 2.6. Behavioral Measures

The behavioral measures were recorded on data collection days (the last 3 days of each period). The buffaloes were video recorded during the daytime from 8:00 to 18:00 h using cameras (Model: FX-C240V Felix, Zetek Communications LLC, Dubai, UAE) attached on the gable walls at 3 m height inside the shed. The buffaloes were individually identified by the numbers assigned to the poles they were tied to. The time spent feeding, standing, and lying were determined by scoring the recorded videos. A buffalo was considered to be feeding when its head and neck were inside the manger area and ended when this criterion was no longer met. A buffalo was recorded as lying when either of the flanks were on the floor [7]. A buffalo not lying was considered as standing.

The meal bout frequency and meal bout length were calculated first by establishing a meal criterion [21] using non-feeding intervals as described by [22]. A meal criterion was the minimum time interval away from the manger such that the next visit was considered as a new meal [21]. The buffaloes were observed exhibiting the behavior of standing in the water tubs during the study time. Standing in the water tub was taken as an indicator of heat stress [23]. A buffalo having either one or both front feet in the tub was considered as an event, and the length of each event was measured.

### 2.7. Statistical Analysis

All the statistical analyses were carried out using SAS (SAS^®^ University Edition: SAS 9.4M6 Institute Inc., Cary, NC, USA). The data collected on individual buffaloes on the last three days of each period were averaged to obtain the period means. The data thus obtained were assessed for normality according to Shapiro–Wilk test using the Univariate Procedure of SAS. The normally distributed data were subjected to ANOVA in a double replicated Latin square design using Mixed Procedure of SAS according to the following model:Y_ijkl_ = µ + T_i_ + S_j_ + P_k:j_ + B_l:j_ + e_ijkl_,
where Y_ijkl_ = the dependent variable; μ = the overall mean; T_i_ = the fixed effect of treatment i, where i = 1, 2, or 3, three sprinkler flow rates; S_j_ = the fixed effect of square j, where j = 1 or 2, June or July; P_k:j_ = the fixed effect of period k within square j, where k = 1, 2, or 3, three periods; B_l:j_ = the random effect of buffalo l within square j, where l = 1 … *n*; and e_ijkl_ = the random error. The Least square means were separated using the PDIFF option with Tukey’s adjusted *p*-values. The differences were considered significant at *p* ≤ 0.05 and a trend at a *p* value between 0.05 and 0.1. The data that were not normally distributed (RR and standing bouts length) were log10-scale transformed and subjected to statistical analysis to determine treatment effects. The means and SEM were back-transformed for presentation purposes. The behavioral data on frequencies (meal, lying, and standing bouts) were descriptively presented as means with SEM because the log transformation did not achieve the condition of approximate normality. The meal criterion was determined plotting the 2-pooled Gaussian distribution of log transformed non-feeding intervals using mixdist [24] package of R (R Core Team, 2020). The standing in the water tub data were analyzed using the Logistic Procedure of SAS, and the association of standing events with treatments was presented as an odds ratio.

## 3. Results

### 3.1. Meteorological Measures

The temperature, relative humidity, and temperature humidity index are summarized in Table 3. The average afternoon shed temperature and THI were 7.6 °C and 5.7 higher than the morning, respectively. The THI and temperature were higher in the evening than in the morning, but lower than in the afternoon. The afternoon outdoor temperature was 4.3 °C higher, and the relative humidity 24.8% lower than inside the shed. A reversed trend was observed for the relative humidity (Table 3). However, both the inside shed and outdoor open area had similar THI in the afternoon (Table 3). The ambient temperature was similar in both the phases (May and June) of the study period, but the RH was relatively higher in June (Figure 2). The THI also had a similar pattern to that of the RH.

### 3.2. Physiological Responses

The sprinkler flow rates significantly affected the physiological responses (Table 4). The buffaloes in the 0.8 L/min sprinkler flow rate group had 0.3 °C higher body temperature (38.7 °C) than in the 1.25 L/min and 2.0 L/min sprinkler flow rate groups (*p* = 0.0041; Table 4). However, the buffaloes in the 1.25 L/min and 2.0 L/min sprinkler flow rate groups had similar body temperature (38.4 vs. 38.4 °C, respectively; SE = 0.06 °C; *p* > 0.05). Similarly, we observed no difference in respiration rate between the 1.25 L/min and 2.0 L/min sprinkler flow rate groups (22.3 vs. 21.4 breaths/min, respectively; SE = 1.03 breaths/min; *p* > 0.05). Whereas, the 0.8 L/min flow rate group had 3.4 and 4.3 more breaths/min (25.7) than the 1.25 L/min and 2.0 L/min sprinkler flow rates groups, respectively (*p* < 0.05; Table 4).

### 3.3. Production Response

The average daily milk yield was 1.1 and 1.7 kg/d higher for the 1.25 L/min (6.3 Kg) and 2.0 L/min (6.6 kg) sprinkler flow rate groups, respectively than the 0.8 L/min group (5.2 Kg; *p* ˂ 0.0001; Table 4). However, we observed no difference between the 1.25 L/min and 2.0 L/min flow rates (6.3 vs. 6.6 kg, respectively; SE = 0.40 kg, *p* > 0.05). The dry matter intake (DMI) was the highest (12.1 kg) for the 2.0 L/min sprinkler flow rate group followed by 1.25 L/min group (11.4 kg) and the 0.8 L/min flow rates (10.5 kg; *p* < 0.0001; Table 4). We also observed that the 0.8 L/min sprinkler flow rate group had 2.4 and 4.4 L more water intake (34.6 L) during the daytime than the 1.25 L/min and 2.0 L/min sprinkler flow rate groups, respectively (*p* < 0.0001), and no significant difference between the 1.25 L/min and 2.0 L/min flow rates (32.2 vs. 30.2 L, respectively; SE = 1.11 L, *p* > 0.05; Table 4).

### 3.4. Behavioral Responses

#### 3.4.1. Feeding Behavior

The sprinkler flow rate treatments did not affect the total feeding time (*p* = 0.6162; Table 5). On average, the buffaloes spent 303.6 ± 10.8 min/10 h in feeding activity. The meal bout frequency and meal bout length were calculated first by establishing a meal criterion. The meal criterion was 28 min. The meal bout length tended to be 14.2 and 14.5 min/10 h higher for the 1.25 L/min and 2.0 L/min sprinkler flow rate groups, respectively, than the 0.8 L/min sprinkler flow rate group (*p* = 0.0762; Table 5). The 1.25 L/min and 2.0 L/min flow rates had similar meal bout length (99.3 vs. 99.6 min/10 h, respectively; SE = 5.28; *p* = 0.0762). In the descriptive statistics, we observed that buffaloes in 0.8 L/min sprinkler flow rate group had 0.4 counts/10 h more meal bouts than the 1.25 L/min and 2.0 L/min flow rates. However, the 1.25 L/min and 2.0 L/min flow rates had the same meal bout frequency (3.8 vs. 3.8 counts/10 h, respectively; SE = 0.15). The data was further explored based on morning and afternoon hours. We observed that sprinkler flow rates 1.25 and 2.0 L/min had significantly longer meal bout length (33.2 and 33.9 min, respectively) in the morning hours than the 0.8 L/min flow rate (*p* = 0.0003; Table 5). Whereas, no significant difference was observed between the 1.25 L/min and 2.0 L/min flow rate groups (138.8 vs. 139.5 min, respectively; SE = 7.9; *p* > 0.05). The descriptive statistics showed that the 0.8 L/min sprinkler flow rate group numerically had the highest meal bout frequency during the morning hours (Table 5).

#### 3.4.2. Lying Behavior

The total lying time ranged from 159.7 to 188.7 (SE = 10.27 min; Table 5). The total lying time and lying bout length tended to be longer be for the 1.25 L/min and 2.0 L/min sprinkler flow rate groups than the 0.8 L/min group (*p* = 0.06; Table 5). The average bout length ranged from 46.3 to 53.0 (SE = 2.29 min; Table 5). Numerically, the average frequency of lying bouts were more in the 2.0 L/min sprinkler flow rate group (3.6 number/10 h) compared to the 0.8 L/min group (3.3/10 h). Similarly, the sprinkler flow rate 2.0 L/min group had more lying bouts both in the morning and afternoon (Table 5).

#### 3.4.3. Standing Behavior

The total standing time of buffaloes in the 0.8 L/min sprinkler flow rate group tended to be longer (440.3 min/10 h) than those in the 2.0 L/min group (411.3 min/10 h; *p* = 0.0621; Table 5). We observed no difference among treatments for standing bout length irrespective of the time of the day (Table 5). The descriptive statistics showed more standing bouts in the 2.0 L/min flow rate group (4.6 and 2.5, overall and morning, respectively) compared to the other treatments (Table 5).

#### 3.4.4. Standing in the Water Tubs

The sprinkler flow rate treatments significantly influenced standing in the water tubs events. The overall odds of standing in water were 80 and 87% less for the buffaloes in the 1.25 L/min and 2.0 L/min sprinkler flow rate groups, respectively, compared to those in the 0.8 L/min group (Table 5). A similar trend was observed among the treatments both for single and multiple standing events per buffalo (Table 6). The average standing duration in water was longer for the 0.8 L/min sprinkler flow rate group (6.4) compared to the 1.25 L/min (3.1) and 2.0 L/min (3.6 min) groups (SE = 0.55 min; *p* = 0.002; Table 5).

## 4. Discussion

### 4.1. Meteorological Measures

In dairy animals, the THI incorporates the effects of the environmental temperature with relative humidity and is commonly used to determine the severity of heat stress [25,26]. The high THI values (84.6) in our study indicated that the buffaloes were under a moderate heat stress [27]. Such high THI values (≥75) have severe negative impacts on the production performance of dairy cows [25]. The threshold level of THI for buffaloes had recently been estimated to be 74 [28]. The high RH inside the shed could be attributed to the showering and reduced air flow compared to the outdoor open area. The difference between the morning and the afternoon temperature and THI values could suggest that buffaloes had less heat load in the morning hours. The higher temperature and THI in the evening than in the morning may serve as a guide to further explore the showering in the evening hours.

### 4.2. Physiological and Production Responses

It is well documented that spray cooling provided either in the holding pen or at the feed bunk is not uncommon, as it lowers body temperature and respiratory rate [5,28]. Compared with the 0.8 L/min sprinkler flow rate, both 1.25 L/min and 2.0 L/min sprinkler flow rates reduced the body temperature and respiration rate of the buffaloes to a similar extent. These sprinkler flow rate treatments (1.25 and 2.0 L/min) could have had more water dripping down the bodies of the buffaloes. Such dripping had been suggested to carry away heat from the animal body [5]. The RT and RR values in the current study were comparable to the buffaloes subjected to showering during summer [10,12]. It could also be possible that relative to the low sprinkler flow rate, both the upper treatments had more water for evaporative cooling, thereby reducing the heat load. Consistent with current findings, a study on dairy cows reported that the water sprinkler flow rates of 1.3 L/min and 4.5 both reduced the body temperature and respiration rate to a similar extent [15].

The buffaloes under the 0.8 L/min sprinkler flow rate had less milk yield, compared to those under the 1.25 L/min and 2.0 L/min sprinkler flow rates, while the 1.25 L/min and 2.0 L/min sprinkler flow rate groups had similar milk yield. In agreement to current findings, the dairy cows cooled with either 1.3 or 4.9 L/min had similar milk yield [7]. Showering had previously showed positive influence on milk yield in buffaloes [13]. About 50% reduction in milk yield under heat stress is attributed to decrease in feed intake [29], and the remaining due to other physiological mechanisms [30]. In this study, both physiological responses did not differ significantly for the 1.25 and 2.0 L/min groups. Hence, the similar milk yield in these groups suggested that the physiological mechanisms of the buffaloes in these two groups were not different. Furthermore, the buffaloes in the 0.8 L/min group were more likely to use energy for maintenance instead of production. Hence, this energy partitioning coupled with the low DMI could justify their lower milk yield compared to the other two groups.

The current findings showed that the DMI decreased with decreased sprinkler flow rates in the buffaloes. A decrease in DMI could be a strategy to reduce the metabolic heat load associated with feeding activity [31]. Contrary to current findings, a study in dairy cows did not find any difference in the DMI with varying levels of sprinkler flow rate [7]. The shorter duration of their study period (2 days) could explain the disagreement between the two findings.

The increased water intake in the 0.8 L/min sprinkler flow rate group indicated that the buffaloes used more water either to compensate for the respiratory water losses due to increased respiration rate or to buffer the rumen under heat stress. A research on heat abatement strategies in buffaloes using fans and showers reported that buffaloes under shower had lower water intake than fans only [13]. In that study, the average water intake was 100.7 L, which was much higher than our study. That difference could be attributed to the monitoring duration of water intake. Their study reported water intake for 24 h, while we measured water intake only during the daytime (10 h/d).

### 4.3. Behavioral Responses

#### 4.3.1. Feeding Behavior

The average feeding time of buffaloes in our study (298.6 to 309.9 min) was in accordance to what had been reported earlier [32]. They reported that the buffaloes spent 309.5 min/24 h in feeding when cooled with sprinklers and fans under shade. In both studies, the buffaloes were fed once daily in the morning and confinement period during the daytime was also similar. That might have explained the similarity in the total feeding time despite the difference of daily monitoring period. Likewise, studies investigating different cooling methods in dairy cows reported feeding time as 5.9 and 5.4 h during 24 h ([6,33], respectively). Slightly higher feeding time in cows could be attributed to milk yield, access to feed, and different housing systems. The higher sprinkler flow rates (1.25 and 2 L/min) had a more positive effect on the feeding behavior during cooler hours compared to the hotter hours of the day. The similar meal bout length across the treatments during afternoon hours indicated that sprinkler flow rates did not influence feeding behavior during hotter hours. However, relative to the 0.8 L/min sprinkler flow rate, the longer meal bouts during morning hours for the 1.25 and 2.0 L/min groups suggested that sprinkler flow rate positively influenced the feeding behavior of the buffaloes during cooler hours of the day.

The shorter meal bout length during the afternoon suggested that a decreased feeding activity of the buffaloes might be a strategy to reduce heat load. The afternoon hours are usually the hottest period of the day [4]. Dairy cows showed a decrease in their feeding time with increased ambient heat load [6]. The lack of difference in the feeding behavior of the 1.25 and 2.0 L/min sprinkler flow rate groups was in agreement to a study in dairy cows where sprinkler flow rates 1.3 and 4.9 L/min did not influence cow behavior [7].

#### 4.3.2. Lying and Standing Behavior

The average time buffaloes spent resting was 174 min in 10 h of the day. This lying time was divided into 50.7 min bouts. A study on heat abatement strategies in buffaloes using fans and showers reported a lying time of 209.2 min/24 h, but did not mention their bout length [32]. Their animals were housed from 600 to 1800 h. Considering the confined duration under shed, the average lying time would be 29.05% that agreed with our findings of 29%. This suggests that they might have also monitored the lying time during the confinement period instead of 24 h. Other studies in cows have reported 12.3 h/d [33], 12.5 h/d [8] divided into 80 min bouts, and 12.1 h/d [7]. These studies showed more lying time. This difference could be explained by the monitoring duration and housing management (tied vs. loose). In our study, the buffaloes were monitored for 10 h (800 to 1800 h) per day inside the pen and the remaining 14 h they were left in the outdoor open area. The buffaloes might have spent more time lying in the open area during nighttime on the relatively comfortable sand bedding. However, further studies covering that period (14 h in the outdoor open area) would provide justified evidence.

Lying time had been reported to decrease in warmer ambient conditions [34,35]. This suggested that buffaloes in the low sprinkler flow rate (0.8 L/min) experienced more heat stress. The increased standing and decreased lying might have enabled them to maximize surface area for sensible and insensible heat loss from their body. It might have also increased the efficiency of respiration [36]. The difference in lying time for the treatments could also be attributed to the differences in the milk yield and the physiological responses. A lying cow has around 5 L/min more blood flow through the udder compared with around 3 L/min for a standing cow [37]. Hence, this might have affected milk production, as had been found in 0.8 L/min sprinkler flow rate group compared to the other two treatments. The results showed no difference in total lying and standing time, and bout length between the 1.25 and 2.0 L/min sprinkler flow rates agreed with previous studies in cows where they found that increasing sprinkler flow rate (1.3 vs. 4.9, 3.3 vs. 4.9 L/min) did not influence cow behavior ([6,7], respectively).

Standing in the water tubs could be a strategy of buffaloes to dissipate heat stress [23] or to get benefits from the wet microclimate [35,36]. The 0.8 L/min sprinkler flow rate might have less water to drip on the concrete floor and thereby less evaporative cooling to affect the floor temperature. Dipping the legs in water could also be a way to avoid the warmer floor surface.

### 4.4. Cooling Efficiency

The current study was the first on water saving strategies to cool buffaloes in a subtropical summer. The average water delivered was 12, 18.75, and 30 L/h per buffalo for sprinkler flow rates 0.8, 1.25, and 2.0 L/min, respectively. The 1.25 and 2.0 L/min sprinkler flow rates cooled buffaloes more effectively than the 0.8 L/min. However, the 1.25 L/min sprinkler flow rate used 37.5% less water and produced a similar cooling effect to that of the 2.0 L/min, as shown in the physiological and production responses. The sprinkler flow rates higher than 1.25 L/min provided little additional heat abatement for buffaloes despite using more water. Hence, the 1.25 L/min sprinkler flow appeared to be more efficient in cooling than the 2.0 L/min. These results agreed with earlier studies where 1.3 L/min sprinkler flow rate was more efficient than the higher flow rates (4.5 and 4.9 L/min; ([6,38], respectively)).

### 4.5. Limitations of the Study

It is important to acknowledge the main limitations of our study. The meteorological measures were only recorded at four time points instead of continuously for 24 h. The 24 h data would have given a much better picture of the intensity of heat loads for the buffaloes. Additionally, we recorded physiological measures once during afternoon hours that limited its use in affirming the efficiency of cooling strategies.

## 5. Conclusions

In comparison to the 0.8 L/min sprinkler flow rate, the 1.25 and 2.0 L/min had lower RT, RR, higher milk yield, and tended to have more lying time. However, the 1.25 L/min flow rate appeared to be more efficient in cooling buffaloes, as it yielded similar production, physiological, and behavioral responses to that of the 2.0 L/min despite using 37.5% less water. The buffalo farmers can devise their showring strategies during the subtropical summer using the current findings.

## Figures and Tables

**Figure 1 animals-11-00339-f001:**
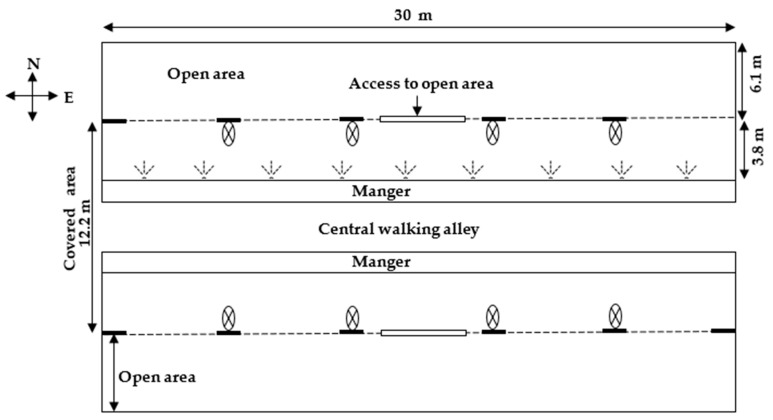
Schematic diagram of the experimental shed to investigate the effect of different sprinkler flow rates and cooling schedules on cooling efficiency in Nili Ravi buffaloes during summer. The dashed lines (---), cross in a circle (
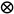
), and combined doted three lines (
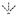
) indicate windows, fans, and sprinklers, respectively, and each sprinkler was for one buffalo.

**Figure 2 animals-11-00339-f002:**
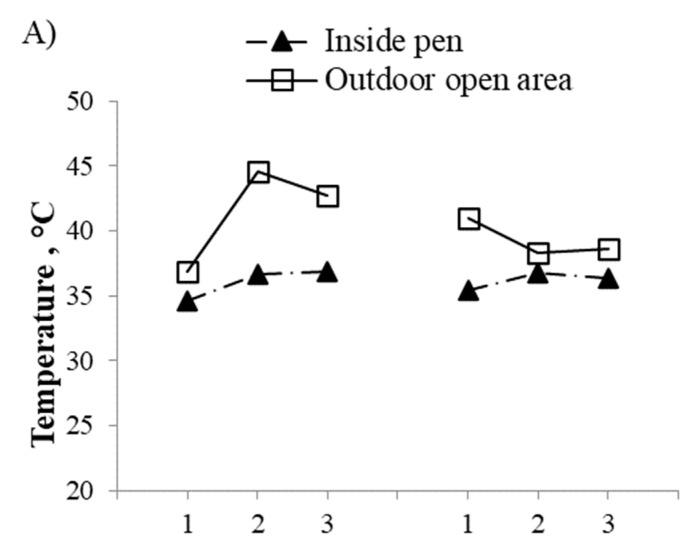
Average temperature (**A**), relative humidity (**B**), and temperature humidity index (THI) (**C**) values for the three study periods in each phase. Shaded triangle represents the inside pen and plain square represents the outdoor open area. Phase 1 lasted for 21 days from 16 May to 5 June and phase 2 from 10 June to 30 June.

**Table 1 animals-11-00339-t001:** Treatment application arrangement in a 3 × 3 Latin square design (*n* = 9).

Period ^1^	Sprinkler Flow Rates ^2^, L/min		
1	0.8	1.25	2.0
2	2.0	0.8	1.25
3	1.25	2.0	0.8

^1^ Each period lasted for 7 days. ^2^ Each sprinkler flow rate was applied to 3 animals in each period.

**Table 2 animals-11-00339-t002:** Water spread characteristics of different sprinkler flow rates.

	Sprinkler Flow Rates, L/min		
Water spread	0.8	1.25	2.0
Along the feed manger, m	0.80	1.31	1.42
Away from the feed manger, m	0.67	0.84	0.88
Covered area, m^2^	0.54	1.10	1.25
Height of nozzles, m	2.3	2.3	2.3
Water use ^1^, L/10 h	120	188	300

^1^ Calculations are based on showering cycle. One cycle consisted of 12 min with 3 min water on followed by 9 min off, making it 15 min shower on in an hour.

**Table 3 animals-11-00339-t003:** Summary of morning and afternoon meteorological measures during the study period (*n* = 146).

Meteorological Measures	Morning ^1^			Afternoon ^2^			Evening ^3^		
	Means	SD	Range	Means	SD	Range	Means	SD	Range
Inside pen
Temperature, °C	28.5	2.8	22.3–32.6	36.1	1.6	33.8–39.0	30.8	3.8	22.7–37.2
Humidity, %	76.3	10.4	59–89	47.9	14.5	23–82	63.4	14.9	46–89
Temperature-humidity index	78.9	4.0	70.7–86.6	84.6	3.3	79.7–91.1	80.2	3.8	71.1–85.9
Outside open area
Temperature, °C	27.6	2.0	24.8–31.4	40.4	3.0	36.4–45.5	35.5	3.8	28.4–39.6
Humidity, %	65.3	12.3	50–95	23.1	10.4	10–39.5	36.3	21.6	10–86
Temperature-humidity index	77.1	2.9	72.5–82.6	84.6	1.5	82.7–87.6	81.9	2.2	78.4–86.2

^1^ Morning category represents measures taken at 6:00 h. ^2^ Afternoon category represents the averages of 13:00 and 15:00 h. ^3^ Evening category represents measures taken at 18:00 h.

**Table 4 animals-11-00339-t004:** Effect of sprinkler flow rate on production and physiological responses of Nili Ravi buffaloes (*n* = 18) during summer.

Variables	Sprinkler Flow Rates, L/min			SEM	*p*-Value
	0.8	1.25	2.0		
Physiological measures ^1^
Rectal Temperature, °C	38.7 ^a^	38.4 ^b^	38.4 ^b^	0.06	0.0041
Respiration Rate, breaths/min	25.7 ^a^	22.3 ^b^	21.4 ^b^	1.03	<0.0001
Production measures
Milk yield, Kg	5.2 ^a^	6.3 ^b^	6.6 ^b^	0.40	<0.0001
Dry matter intake (DMI), Kg	10.5 ^a^	11.5 ^b^	12.1 ^c^	0.36	<0.0001
Water intake, L	34.6 ^a^	32.2 ^b^	30.2 ^b^	1.11	<0.0001

^a–c^ Values with different superscripts in a row are significantly different (*p* < 0.05). ^1^ Recordings were taken between 1300 to 1400 h.

**Table 5 animals-11-00339-t005:** Effect of sprinkler flow rate on behavioral responses of Nili Ravi buffaloes (*n* = 18) during summer.

Variables	Sprinkler Flow Rates, L/min				*p*-Value
	0.8	1.25	2.0	SEM	
Feeding behavior/10 h
Total feeding time ^1^, min	298.6	309.9	302.4	10.8	0.6162
Meal bout length ^2^, min/bout
Overall	85.1	99.3	99.6	5.28	0.0762
Morning ^3^	105.6 ^a^	138.8 ^b^	139.5 ^b^	7.9	0.0003
Afternoon	51.7	51.8	51.0	2.84	0.9606
Meal bout frequency, No.
Overall	4.2	3.8	3.8	0.15
Morning	1.3	1.2	1.1	0.06
Afternoon	2.9	2.9	3.0	0.14
Standing behavior/10 h
Total standing time ^4^, min	440.3	426.4	411.3	10.26	0.0612
Standing bout length ^5^, min/bout
Overall	99.8	99.9	89.6	1.06	0.1236
Morning	90.7	92.7	81.9	1.09	0.4691
Afternoon	109.9	108.3	98.0	1.06	0.2236
Standing bouts frequency, No.
Overall	4.3	4.2	4.6	0.14
Morning	2.1	2.0	2.1	0.09
Afternoon	2.3	2.2	2.5	0.09
Lying behavior/10 h
Total lying time ^6^, min	159.7	173.6	188.7	10.27	0.0618
Lying bout length ^5^, min/bout
Overall	46.3	52.9	53.0	2.29	0.0636
Morning	52.7	63.3	60.8	3.61	0.1169
Afternoon	40.2	42.7	45.2	2.33	0.2609
Lying bout frequency, No.
Total	3.3	3.2	3.6	0.15
Morning	1.7	1.5	1.8	0.09
Afternoon	1.8	1.9	1.9	0.10
Standing in water tub ^7,^ min	6.4 ^a^	3.1 ^b^	3.6 ^b^	0.55	<0.0001

^a–b^ Values with different superscripts in a row are significantly different (*p* ≤ 0.05). ^1^ A buffalo was considered to be feeding when its head and neck were inside the manger area and ended when this criterion was no longer met. ^2^ A meal bout is the sum of the feeding and non-feeding intervals based on the meal criterion. ^3^ Daytime was classified into two categories; morning (800 to 1159 h) and afternoon (1200 to 1800 h). ^4^ A buffalo not lying was considered as standing either feeding or not. ^5^ Continuous duration of standing or lying. ^6^ A buffalo was recorded as lying when either of the flanks were on the floor. ^7^ When buffaloes put either one or both front legs inside their water tubs.

**Table 6 animals-11-00339-t006:** Association of sprinkler flow rate with the standing in water tubs behavior of Nili Ravi buffaloes (*n* = 18).

Standing in Water Tub	Sprinkler Flow Rates (L/min)	Odds Ratio ^1^	95% CI ^2^
Overall standing events	0.8 1.25	1.0 0.202 ^4^	Ref ^3^ 0.098–0.416
	2.0	0.129	0.059–0.283
Single events per buffalo ^5^	0.8 1.25	1.0 0.258	Ref 0.118–0.563
	2.0	0.151	0.063–0.359
Multiple events per buffalo ^6^	0.8 1.25	1.0 0.076	Ref 0.098–0.416
	2.0	0.046	0.059–0.283

^1^ Odds ratio = odds of standing in water tub. ^2^ If 95% CI does not include 1, it indicates statistically significant difference at *p* < 0.05. ^3^ Reference group. ^4^ Example interpretation: Buffaloes in 1.25 sprinkler flow rate group had 0.20 times lesser (80% less) odds of standing in water tubs. ^5^ A buffalo exhibiting standing in water tub behavior only once during a recording session (800–1800 h). ^6^ A buffalo exhibiting standing in water tub behavior more than once during a recording session (800–1800 h).

## Data Availability

The data presented in this study are available on request from the corresponding author.

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
