# Peer review of "Effects of Sprinkler Flow Rate on Physiological, Behavioral and Production Responses of Nili Ravi Buffaloes during Subtropical Summer"

_animals, 2021, doi:10.3390/ani11020339_

Round 1

Reviewer 1 Report

The article addresses an important issue, especially for countries with a tropical or subtropical climate, where issues of thermal stress play an important role in animal production.
The thermal cooling of animals using sprinklers is a technique already widely used, and this work aims to improve its efficiency, not only from the perspective of animals but also from the perspective of the efficiency of the water resource (through different flows rates).

The text is written in a clear and simple way, which makes reading quick and easy to understand.

Materials and methods
The use of the temperature and humidity index (THI), as the authors refer, is commonly used. However, it does not mean that it is the most suitable in bioclimatology studies. The black globe temperature and humidity index (BGTHI) gives us a more realistic perspective of what is really the situation of thermal stress that animals present because it considers the effect of the wind. The present article would be more complete if it used BGTHI, being able to even compare the two indices (BGTHI and THI).

Results
Meteorological measurements can only be shown in the table since the information given by the graphs does not add any information.
However, in the case of behavioural responses, the presentation of the results in the form of a graph (area chart) would allow the reader to get a perception of the animals' behaviour throughout the observation period, in the differents studied sprinkler flow rates.

Discussion
Studies on heat stress in dairy animals are numerous and, therefore, I believe that the authors will be able to enrich the work if they include more references.
In addition, the discussion of the results obtained may be more detailed, especially in the case of variables whose results were not significantly different but which show a tendency towards.

Author Response

We are grateful for allowing us to submit a revised version of this article. The comments of the reviewers greatly helped us in improving the quality of the manuscript. We have corrected the grammatical errors especially the use of “articles”, removed the repetition in discussion, and improved the clarity of the results as per the reviewers’ suggestions. Further, to guide the reviewers the modified parts of the manuscript are highlighted as yellow. The specific comments of the reviewers have been addressed as an attached file

Reviewer 2 Report

The research focuses on a critical and timely issue - ensuring welfare and responsbile resource use while finding a more sustainable solution. I think overall the paper has merit but there are some details missing, some grammatical challenges, repetition in the discussion, and clarity/conciseness needed in the results. Please see specific comments below:

The formatting (large indent, font with subtitles, etc) is incorrect but I assume that will get fixed during the editing process. The paper needs to be revised for grammar and punctuation. I identified some of the instances but it became too burdensome to edit so I wanted to make clear that the grammatical revisions listed are not the only ones that should be addressed. Often an “a” or “the” is missing in front of a noun. Related to the research question, why is potable water used instead of non-potable water? That may help to include somewhere as the potable nature of the water seems to be a big concern.

Simple Summary: this is not a complete simple summary. It should also include a sentence or two about background and why the study was necessary/important.

Abstract: the abstract limit is 200 words. This needs to be significantly reduced – it is currently at 414. “Riverine” is used in the abstract but nowhere else in the paper. If you want to use this word it should also be introduced in the paper.

Keywords : should be in alphabetical order

Introduction

Line 45: insert ‘the’ before Indo-Pak

Line 46: Instead of ‘they’ say “Water buffaloes”

Line 50: insert “has” before “led”; The sentence beginning with “Such scenario” is missing some words, adjust.

Line 53: missing period

Line 56: The way this sentence is written it suggests you are saying “physiological performance” – what is an improvement in physiological performance. Clarify.

Line 57: insert ‘a’ before ‘large’

I stopped editing grammar at this point – this needs to be addressed throughout even though I will not provide specific edits from this point forward.

Line 59-62: need to cite your survey as either unpublished data or personal communication – something to indicate where you got this information.

Line 66-67: need a citation

Line 68: do not start a sentence with ‘But’

It would be helpful to explain generally how this water is applied. One animal in a pen? Group pen? A few sentences about the housing system and the watering system would help readers that are unfamiliar.

Materials and Methods:

General comment: How were the treatments dispersed in the barn so as not to have one treatment impact the cooling of an animal in another treatment?

Figure 1. – add dimensions. Additionally, how far apart were the animals from each other? Evenly spaced in this diagram?

Line 85-86: define DIM; the list of the parameters is also not the best presentation – consider presenting differently; include unit for bodyweight; and you do not need respectively.

Line 87: what is face to face housing?

Line 100: why are they tethered in the day and not at night?

Table 2. Remove first ‘area’ from “area covered area”

Line 140-1: What was 1800 categorized as?

Line 147: change “at” to “between”

Line 147-8: Diet only adds up to 99%

Line 154: What is “daytime” describing? Daytime hours?

Line 156: What time was the remaining water measured?

Line 159: was milk yield recorded at each milking?

Line 164: “humanly” should be “humanely”; what does humanely restrained mean?

Behavioral measures – your recording and sampling rules are not clear – please clarify.

Line 173: What are the data collection days? Not clear what those are vs non data collection days.

Line 175: “both” implies there are two things the cameras were attached to but you only say gable walls – remove both or clarify

Line 182-4: The “meal criterion” definition and description is unclear. Please rewrite this part to provide a clear explanation of what this means.

Line 184-6: This does not belong in this section – should be in statistical analysis description.

Line 187-8: What does this sentence mean?

Figure 2. should be in the results section. The different lines and colors need to be explained. The significance and meaning of this graph is not clear. Need to explain meal criterion.

Line 200: add an additional sentence to explain in a more clear way your collapsing of data. This is a key element so you need to ensure readers understand

Line 201: “approximately normally?”

The presentation and formatting of the equation should be clarified.

Line 220-1: What about the 1800h measurement?

Results – The description of results that only showed a tendency can be reduced and made more concise. All table and figure titles/legends should include the total n included in the data presented.

Section 3.1 – Results paragraph needs to be written more clearly. For example -Line 223-3: You list 7.6 as both the temp and THI which is not correct – they are not the same value and there is no 7.6 as a value. Or if you are saying 7.6 x greater the sentence is not written correctly. Inside or outside temperature?

Figure 3 should include time frames of each period.

Section 3.2. The main effect of sprinkler flow rate for each outcome should also be stated in the text, not just the table.

Table 4. Physiological parameters should be before production parameters in the table as they are discussed in that order in the text; need to include the n in the table

Line 289-91: This sentence is unclear – placement of numbers makes it unclear.

Table 5. All behaviors should be defined in the footnotes, not just standing in the water tub.

Table 6. What are single and multiple standing events?

Discussion – somewhere in the paper, maybe in the beginning of the discussion, you need to define what you determine as “more efficient”. What makes the sprinkler rates more efficient than another? Consider removing/condensing some subheadings in the discussion – currently the subheadings have resulted in considerable repetition between sections, in the first paragraphs – some examples noted below. This would also enable you to relate the physiological impacts with the production impacts. How do the benefits you see of treatments on physiological parameters influence/relate to the production parameters? The Discussion would be more effective with fewer subheadings and a more fluid narrative.

Line 341: equivalent responses is not the appropriate language to use.

Line 347-8: Although this is relevant here, this same info is stated in the previous paragraph

Line 348: “severe” not “sever”

Line 350-4: Is this necessary? By describing THI you address this.

Line 359-61. This sentences is very similar to the first sentence of the discussion – repetitive.

Line 361-2; the meaning of this sentence is unclear.

Line 370: type – showring

Line 374: “almost a similar extent” is not clear and almost contradictory

Line 379: similar milk yield in the cited study between treatments? Or similar to your study?

Line 390-1: “due to ethical concerns” seems unnecessary

Section 4.4.2. Consider dividing this section into paragraphs

Line 433-4: “going by” is a bit slangy

Subtitle 4.6 – missing a word? Improve this subtitle

Line 485-7: What does this mean? And what is its relevance? Needed?

Line 488: No future research directions were discussed so to include this some need to mentioned. Additionally, this sentence is not a limitation as stated in the subheading title.

Conclusion:

I think there needs to be a few more sentences explaining the application of this research. It would also be beneficial to include the metrics used to determine “effective cooling”, i.e., what factors showed the diferent rates were affected.

Line 495: “cooling” not “colling”

Acknowledgements – this section includes just the template text.

Author Response

We are grateful for allowing us to submit a revised version of this article. The comments of the reviewers greatly helped us in improving the quality of the manuscript. We have corrected the grammatical errors especially the use of “articles”, removed the repetition in discussion, and improved the clarity of the results as per the reviewers’ suggestions. Further, to guide the reviewers the modified parts of the manuscript are highlighted as yellow. The specific comments of the reviewers have been addressed as follows.

Round 2

Reviewer 2 Report

Thank you for addressing all of the comments. The manuscript is much improved.

I think everything has been addressed adequately except Figure 3. The explanation, presentation and discussion of meal criterion is not clear. Figure 3 has not been adequately explained. The way it is currently presented, it is not clear the value of including this figure. I would suggest making this more clear and demonstrating the value of its inclusion or removing it from the mansucript. 

Author Response

Dear Reviewer,

We greatly appreciate the comments. We have addressed the specific comments in the attached file please.  
